# IVQA-LD: Inclusive Multimodal Understanding for Population with Limb Deficiency

**Yan Ke** [1]  **Xin Shen** [1]  **Jiaying Ying** [1]  **Xin Li** [2]  **Xin Yu** [3]

## Abstract

People with limb differences often face significant challenges in accessing inclusive AI services, largely due to the lack of structured, high-quality resources centered on disability contexts. In this work, we introduce a limb-deficiency aware body-centric learning and evaluation paradigm that involves (i) a large-scale limb-aware vision–language dataset and evaluation benchmark for multimodal reasoning, and (ii) a model adaptation strategy for Vision-Language Models (VLM) in limb-difference contexts. Specifically, we first collect limb-difference data covering all eight limb-deficiency types across diverse real-world scenarios. The data are systematically organized into 96 limb-affected human action categories and 68 medical-functional classes defined by the World Health Organization (WHO). Then, we curate an expert-annotated vision–language dataset for limb-aware multimodal understanding, named Inclusive VQA for Limb Deficiency (**IVQA-LD**). IVQA-LD comprises 80K VQA pairs spanning eight core tasks including visual grounding, quantitative reasoning, functional semantic classification, and instructional text generation. We benchmark state-of-the-art VLMs on IVQA-LD and find that they consistently struggle across all tasks, exposing substantial deficiencies in limb-aware perception and reasoning. To address this, we further propose a Body-centric Structure-aware Initialization (BSI) strategy that aligns model representations with limb-specific semantics. With BSI, VLMs fine-tuned on IVQA-LD achieve significant performance improvements across all the tasks. We will publicly release the dataset to support future research. The code and data will be available at ⚡ IVQA-LD.

[1]The University of Queensland, Brisbane, Australia [2]Macquarie University, Sydney, Australia [3]Adelaide University, Adelaide, Australia. Correspondence to: Xin Yu <xin.yu@adelaide.edu.au>.

*Proceedings of the 43rd International Conference on Machine Learning*, Seoul, South Korea. PMLR 306, 2026. Copyright 2026 by the author(s).

## 1. Introduction

Vision–language models (VLMs) have achieved rapid progress in recent years, demonstrating strong multimodal reasoning and generalization across diverse visual understanding tasks (Radford et al., 2021; Li et al., 2022). However, concerns regarding fairness and representation in training data have become increasingly prominent (Garcia et al., 2023). Among underrepresented populations, people with limb differences remain largely absent from existing large-scale vision datasets. Despite the fact that hundreds of millions of individuals worldwide live with acquired or congenital limb loss (Ying et al., 2025; Du et al., 2026), mainstream VQA datasets rarely include limb-deficient individuals and seldom capture prosthetic usage or adaptive human actions. As a result, current VLMs often fail to accurately detect missing limbs, identify prosthetic devices, or reason about functional capabilities, leading to systematic errors and inequitable model behaviors (Cao et al., 2025; Shen et al., 2025b).

Recent efforts have begun exploring inclusive perception, such as assistive datasets targeting blind and low-vision users (*e.g.*, VizWiz (Gurari et al., 2018) and mmWalk (Ying et al., 2026)), deaf and hard-of-hearing individuals (Li et al., 2020; Shen et al., 2023; 2024; 2026), and pose estimation datasets focusing on individuals with limb deficiency (Ying et al., 2025; Huang et al., 2024; Du et al., 2026; Ying et al., 2024). However, these datasets do not support multimodal reasoning about limb differences, nor do they provide fine-grained semantic annotations grounded in medical-functional knowledge or body-centric understanding. To date, the community lacks a benchmark that systematically evaluates VLMs on limb-deficiency-aware multimodal understanding and reasoning.

To bridge this gap, we introduce a limb-deficiency-aware body-centric benchmark for multimodal learning and evaluation. As the foundation of this benchmark, we curate a large-scale limb-aware vision-language dataset, termed Inclusive Visual Question Answering for Limb Deficiency (**IVQA-LD**). IVQA-LD covers diverse real-world scenarios, including daily living activities, rehabilitation scenarios, and

Paralympic sports.[1] The dataset spans all eight categories of limb deficiency, 96 limb-affected human actions, and 68 medical-functional classes defined by the World Health Organization (WHO) (World Health Organization, 2025).

In IVQA-LD, we ask multiple annotators to label generic image attributes that do not require expert knowledge, including bounding-box annotations for residual limbs, labels indicating the presence of prosthetic or assistive devices, spatial localization of these devices in images, and whether a person is using assistive devices. More importantly, to capture clinically meaningful semantics, biomechanics experts and rehabilitation professionals are invited to design question-answer (Q-A) pairs focusing on limb-aware and function-centric concepts, such as residual limb identification, prosthetic function, adaptive actions, and rehabilitation guidance. To ensure annotation quality, all machine-generated Q-A pairs are fully reviewed by domain experts, and the human-authored subset is further validated through random expert cross-review. Only samples validated or refined through expert consensus are retained. Based on these expert annotations, IVQA-LD supports eight limb-aware core tasks, including limb-aware visual classification, functioning and disability classification, action recognition, prosthesis counting, visual grounding, assistive device and prosthesis–human interaction understanding, scene reasoning, descriptive caption and rehabilitation-specific instructional guide generation. In total, IVQA-LD contains 80,000 image Q-A pairs derived from 14,054 images.

Leveraging our IVQA-LD, we conduct the first comprehensive evaluation of state-of-the-art VLMs, including both open-source and closed-source models, in limb-deficiency contexts. Our experimental results demonstrate that existing models struggle across nearly all the tasks. Common failure patterns include misidentifying prosthetic limbs, hallucinating intact limbs, and producing vague or clinically incorrect guidance. These findings reveal that current VLMs lack the body-centric structural understanding required for accurate limb-aware perception and reasoning.

To address this limitation, we propose a *Body-centric Structure-aware Initialization (BSI) strategy* that explicitly incorporates limb- and body-structural priors during fine-tuning. Our BSI strategy is inspired by how non-experts gradually acquire domain expertise when reasoning about limb deficiency. Specifically, we first warm up VLMs using limb-related appearance cues that only require general limb-deficiency knowledge, such as bounding boxes of residual limbs and spatial locations of prosthetic or assistive devices. This enables VLMs to acquire sufficient knowledge on residual limbs, prosthetics and assistive devices. Afterwards, we fine-tune the models on IVQA-LD by jointly supervising limb-related localization and disability-specific question an-

swering, rather than relying solely on question answering supervision. This training strategy encourages VLMs to attend to limb-deficiency-relevant regions before generating answers, and mitigates language-only shortcut learning as well as superficial question–answer correlations, leading to significant performance improvements. Consequently, fine-tuning with BSI improves performance across all eight core tasks, for example achieving over 50% accuracy on visual grounding and 73% on functional and disability classification. These results underscore the effectiveness of structured domain knowledge transfer when adapting general-purpose VLMs to disability-centered applications.

Our contributions are summarized as follows:

- We introduce IVQA-LD, the first large-scale expert-annotated limb-deficiency-related visual question answering dataset that supports, for the first time, limb-deficiency-aware multimodal learning and evaluation.

- We conduct the first comprehensive benchmark evaluation of state-of-the-art VLMs in limb-deficiency contexts, revealing consistent failure patterns in limb-aware perception and reasoning.

- We propose a Structure-Aware Initialization strategy that incorporates limb- and body-structural priors during fine-tuning, significantly improving multimodal understanding in limb-deficiency scenarios.

## 2. Related Work

### 2.1. Multimodal Large Language Models

Recent progress in multimodal understanding has been driven by Multimodal Large Language Models (MLLMs), which combine vision encoders with Large Language Models (LLMs) to enable joint visual–linguistic reasoning. Early models such as BLIP-2 (Li et al., 2023) and LLaVA (Liu et al., 2023) established a common architecture that connects a pretrained vision encoder (e.g., CLIP) to a pretrained LLM (*e.g.*, Vicuna or LLaMA (Grattafiori & Llama team, 2024)) via a lightweight trainable projection module. Through visual instruction tuning, these models can follow open-ended instructions and generalize across a wide range of tasks.

LLaVA-NeXT (Liu et al., 2024a) improves performance through higher-resolution visual inputs and curated training data. The Qwen-VL series (Bai et al., 2023; Wang et al., 2024) and InternVL (Chen et al., 2024) further extend model capacity and training scale, achieving strong results on general-purpose VQA and captioning benchmarks.

Despite their success, most MLLMs are trained predominantly on broad, web-scale multimodal data and evaluation benchmarks derived from general visual domains. They

---

[1]All the data are licensed for research purposes.

*Table 1.* Comparison of IVQA-LD with representative general-purpose and expert-domain VQA datasets.

| Dataset | Domain | Image Source | Annotation | # Q-A Pairs | Key Tasks / Focus |
|---------|--------|--------------|------------|-------------|-------------------|
| VQA v2 (Goyal et al., 2017) | General | COCO | Crowdsourced | 1.1M | Visual recognition, Counting |
| GQA (Hudson & Manning, 2019) | General | Visual Genome | Programmatic | 22M | Compositional reasoning |
| OK-VQA (Marino et al., 2019) | General | COCO | Crowdsourced | 14K | External knowledge reasoning |
| VizWiz-VQA (Gurari et al., 2018) | Assistive (Visual) | Real-world (Users) | Crowdsourced | 31K+ | Assistive QA, Unanswerable queries |
| VQA-RAD (Lau et al., 2018) | Medical | MedPix | Expert | 3.5K | Clinical reasoning |
| SLAKE (Liu et al., 2021) | Medical | Multi-source | Expert | 14K | Multimodal clinical QA |
| **IVQA-LD (Ours)** | **Assistive (Limb)** | **Real-world** | **Disability-Expert** | **80K** | **Visual, Functional, Rehabilitation** |

often lack fine-grained, domain-specific understanding capabilities, particularly in limb-deficient functional capability and assistive reasoning. This limitation motivates the need for dedicated benchmarks that evaluate and improve limb-aware body-centric multimodal understanding.

### 2.2. Expert-Driven and Assistive VQA Datasets

To support domain-specific reasoning beyond generic VQA, several expert-driven datasets have been introduced. In the medical domain, datasets, such as VQA-RAD (Lau et al., 2018) and SLAKE (Liu et al., 2021), focus on radiology and clinical imagery, requiring models to answer expert-formulated questions based on X-ray, CT, or MRI scans. These benchmarks emphasize specialized knowledge and structured reasoning rather than everyday visual concepts.

In assistive technology, VizWiz-VQA (Gurari et al., 2018) is a prominent benchmark designed to reflect real-world challenges faced by blind and low-vision users. It contains images captured by users in unconstrained environments, paired with spoken questions, and also includes issues such as image blur, occlusion, and unanswerable queries. VizWiz-VQA has played a key role in advancing assistive VQA research (Chen et al., 2022).

As shown in Table 1, existing assistive VQA datasets primarily focus on visual impairment, while expert-annotated datasets are largely limited to internal medical imagery. In contrast, IVQA-LD targets external, real-world, body-centric scenarios involving physical limb differences, combining expert annotations with diverse daily activities. This distinguishes it from prior benchmarks and enables systematic evaluation of limb-aware multimodal reasoning.

### 2.3. Inclusivity-aware Dataset

Large-scale vision and multimodal models trained on un-curated web data have been shown to learn and amplify societal biases, motivating extensive research on fairness and inclusivity in computer vision (Dehdashtian et al., 2024). More recently, as MLLMs have become widely deployed, researchers have begun to systematically evaluate their demographic and representational biases using dedicated benchmarks such as FACET (Wu et al., 2025). These studies reveal persistent performance gaps across different popula-

tion groups, underscoring the need for targeted datasets and evaluation protocols that better represent underrepresented populations. LDPose (Ying et al., 2025), the first pose estimation dataset focusing on individuals with limb deficiency, demonstrates that existing pose estimation models perform poorly on subjects with limb differences than on able-bodied individuals. This finding highlights the broader limitations of current vision models when applied to limb-deficiency contexts. Driven by this, we curate a vision-language dataset to study fairness and reliability of MLLMs in understanding and reasoning about populations with limb deficiencies.

## 3. Limb-deficiency-aware Multimodal Learning and Benchmarking

### 3.1. Data Source

Our IVQA-LD is constructed from the Limb-Deficient Pose (LDPose) collection (Ying et al., 2025), which provides a large number of images depicting individuals with limb deficiencies. LDPose aggregates images from publicly available visual media and is curated to cover a wide range of daily-life and athletic scenarios, making it a suitable foundation for limb-deficiency-aware multimodal analysis.

We found that although LDPose contains approximately 28K images, several images are extracted from the same video footage, include able-bodied individuals, or exhibit insufficient resolution. Since our goal is to enable VLMs to attend to limb-deficiency-relevant regions rather than perform pose estimation on all individuals in an image, we apply a series of filtering steps to select representative, high-quality images of individuals with limb deficiency. Specifically, we first remove able-bodied instances using the ground-truth pose annotations provided in LDPose. We then manually discard images in which residual limbs are heavily occluded or not clearly observable, as well as near-duplicate frames originating from the same video footage. These filtering criteria ensure that the remaining images support reliable downstream annotations and provide informative supervision for residual-limb bounding boxes. After filtering, we retain 14,054 high-quality images in which all target individuals are clearly visible at sufficient resolution.

Considering the physical, functional, and assistive-device

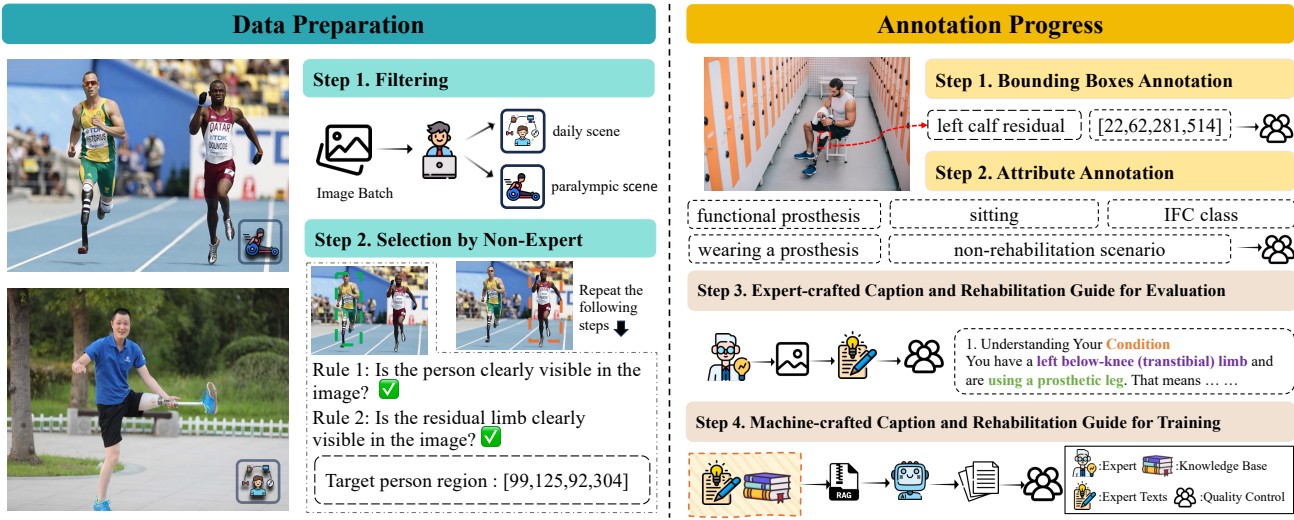

Figure 1. Overview of our IVQA-LD curation pipeline. Left: data filtering and selection process. Right: data annotation process combining non-expert annotators with domain experts.

differences between elite Paralympic athletes and individuals with limb deficiency (Tinney et al., 2024), we further categorize the images into two groups: daily-life scenes and Paralympic scenes, as illustrated in Figure 1. This categorization facilitates the design of context-aware Q-A pairs that account for differences between routine activities and competitive athletic environments.

### 3.2. Limb-related Generic Image Annotations

To support structure-aware multimodal learning, we first collect a set of limb-related generic image annotations that do not require domain-specific expertise. These annotations provide explicit visual and contextual cues, such as the locations and presence of residual limbs, prosthetics, and assistive devices, which guide VLMs to attend to limb-deficiency-relevant regions in learning. Although these annotations do not inherently require expert-level knowledge, biomechanics experts and rehabilitation professionals are still invited to provide training to annotators, particularly in recognizing prosthetic and assistive devices, in order to ensure annotation consistency and reliability.

**Residual limb bounding boxes:** We begin the annotation process by localizing all residual limbs in each image. We observe that VLMs often overlook residual limb cues, particularly when such cues occupy small regions or are visually indistinct. Thus, we introduce bounding-box annotations as explicit spatial supervision to highlight limb-deficiency-relevant regions.

Seven trained annotators manually label approximately 14K images using an eight-class fine-grained taxonomy, covering left/right upper-arm residuals, left/right forearm residuals, left/right thigh residuals, and left/right calf residuals,

as shown in Figure 1. These annotations provide precise supervision of residual limb locations and types, forming the basis for limb-aware localization tasks and supporting body-centric structure-aware initialization during training.

**Attribute annotations:** Beyond spatial localization of residual limbs, each target individual is further annotated with a set of body-centric attributes that capture functional and contextual information relevant to limb-deficiency understanding. Specifically, we annotate (a) assistive device type, including "crutch", "wheelchair", "walker", "simple walking aid", "functional prosthesis", "sports prosthesis", "cosmetic prosthesis", (b) prosthesis type, including "functional prosthesis", "sports prosthesis", "cosmetic prosthesis", (c) indicator of prosthesis or assistive device usage, (d) number of prosthetic or assistive devices, (e) scene type (*i.e.*, daily or Paralympic scenes), and (f) action class, such as "wear clothes", "Prosthesis Donning", "Assistive Device Doffing" and "javelin throw". These attributes provide semantic signals that support higher-level reasoning tasks such as functional capability assessment, action understanding, and human-assistive device interaction analysis.

**Quality control of non-expert annotations:** In the annotation process, each annotator is assigned a batch of 100 images each round. To control bounding-box annotation quality, 5% of bounding-boxes from each batch are randomly selected and independently re-annotated by a second annotator. We then compute the mean Intersection-over-Union (mIoU) between the two sets of annotations. If the mIoU falls below 80%, the entire batch is flagged and re-assigned for re-annotation by a third annotator. A similar quality control protocol is applied to attribute annotations. Each annotator labels attributes for batches of 100 images,

from which 5% of the samples are randomly selected for independent verification by another annotator. If more than 5 attributes (*e.g.*, numerical counts or categorical labels) are found to be incorrect, the corresponding batch is flagged and reassigned for re-annotation by a third annotator.

### 3.3. Limb-deficiency Domain-specific Annotations

While generic annotations provide essential visual attributes, limb-deficiency-aware multimodal reasoning also requires clinically meaningful semantics. Therefore, we collaborate with two rehabilitation specialists and two biomechanics experts to provide limb-deficiency domain-specific attribute annotations and VQA pairs.

**Disability classification:** There are two distinct scenarios based on scene types: (1) in daily-life scenes, experts adopt the World Health Organization (WHO) International Classification of Functioning, Disability and Health (ICF) (World Health Organization, 2025), where codes such as s730 (upper limb) and s750 (lower limb) denote affected body structures and subsequent digits specify severity and anatomical location; (2) in sports scenes, experts adopt the World Para Athletics (WPA) classification system (International Paralympic Committee, 2023), which reflects functional eligibility criteria in competitive Paralympic settings. This dual-standard design ensures that disability annotations remain both medically grounded and contextually appropriate.

**VQA types and generation:** IVQA-LD defines a comprehensive taxonomy of VQA tasks, structured into **eight** question types across **four** levels: (1) *Perception*: visual grounding; (2) *Classification*: limb-aware visual classification, assistive device and prosthesis–human interaction understanding; (3) *Reasoning*: prosthesis counting, action recognition, scene reasining, functioning and disability classification; (4) *Generation*: descriptive caption and instructional guide generation. Questions and ground-truth answers are authored or reviewed by domain experts to ensure clinical relevance and functional correctness. Distractor options are automatically generated by GPT-5 via carefully designed prompts to produce answer choices that are similar in format and semantically plausible yet incorrect. As illustrated in Figure 2(e), this taxonomy captures a broad spectrum of limb-aware multimodal understanding tasks, while Figure 2(a–d) further summarize the distributions of representative functional classes, deficiency types, action categories, and functional classifications.

**Expert-generated captions and guides.** We invite rehabilitation specialists to provide expert-authored image captions and rehabilitation guidance for a subset of daily-life images. Specifically, 1,500 images are selected based on their representativeness and clinical relevance as assessed by domain experts. For these images, experts produce 1,500 descriptive captions and 1,000 rehabilitation guidance annotations.

The expert descriptions emphasize functional interpretation, compensatory strategies, and assistive device usage, ensuring both medical and biomechanical accuracy. To improve annotation efficiency while maintaining quality, experts provide captions and guidance through voice recordings, which are subsequently transcribed into text using automatic speech recognition software Whisper (Radford et al., 2023). This workflow significantly reduces annotation overhead, allowing each expert to complete the assigned annotations in under 10 minutes per image on average. Building on these expert-authored texts and external domain knowledge, we further employ a retrieval-augmented generation (RAG) pipeline (Lewis et al., 2020) to synthesize 11.3K captioning and rehabilitation guidance Q-A pairs for training.

**Quality control of expert annotations:** To ensure annotation reliability, all the manually created and automatically generated Q-A pairs undergo expert review to verify answer correctness and uniqueness. For the human-authored subset, which contains 2,500 expert-curated Q-A pairs for captions and rehabilitation guidance, we adopt a sampling-based audit strategy. Specifically, 15% of this subset, corresponding to 375 Q-A pairs, is randomly sampled for expert cross-review. Meanwhile, all machine-generated Q-A pairs, including 11.3K samples, are fully reviewed by experts. If any sampled human-authored Q-A pair exhibits inconsistent or ambiguous answers, it is jointly reviewed and refined by two domain experts. Machine-generated Q-A pairs that are found to be incorrect or clinically implausible are discarded. In practice, the vast majority of manually created Q-A pairs require little to no revision during the quality control process.

### 3.4. Data Statistics of IVQA-LD

Overall, our IVQA-LD dataset contains 80,000 visual Q-A pairs, covering 8 limb-deficiency categories, 96 limb-affected human actions, and 68 functional categories. This diverse composition provides a challenging and inclusive benchmark for limb-aware vision-language understanding. IVQA-LD spans four levels of tasks as mentioned in Sec. 3.3. The distribution of Q-A types across these task categories is reported in Figure 2.

Regarding annotation sources, approximately 90% of the VQA pairs are generated by trained non-expert annotators following structured annotation guidelines for objective visual tasks, while annotations requiring clinical or rehabilitation-specific reasoning are authored or fully verified by domain experts, including rehabilitation and biomechanics experts. This design balances annotation scalability with clinical and functional reliability.

For benchmarking and model evaluation, we split the dataset into training, validation, and test sets following a **7:1:2** ratio. The split is performed independently within each

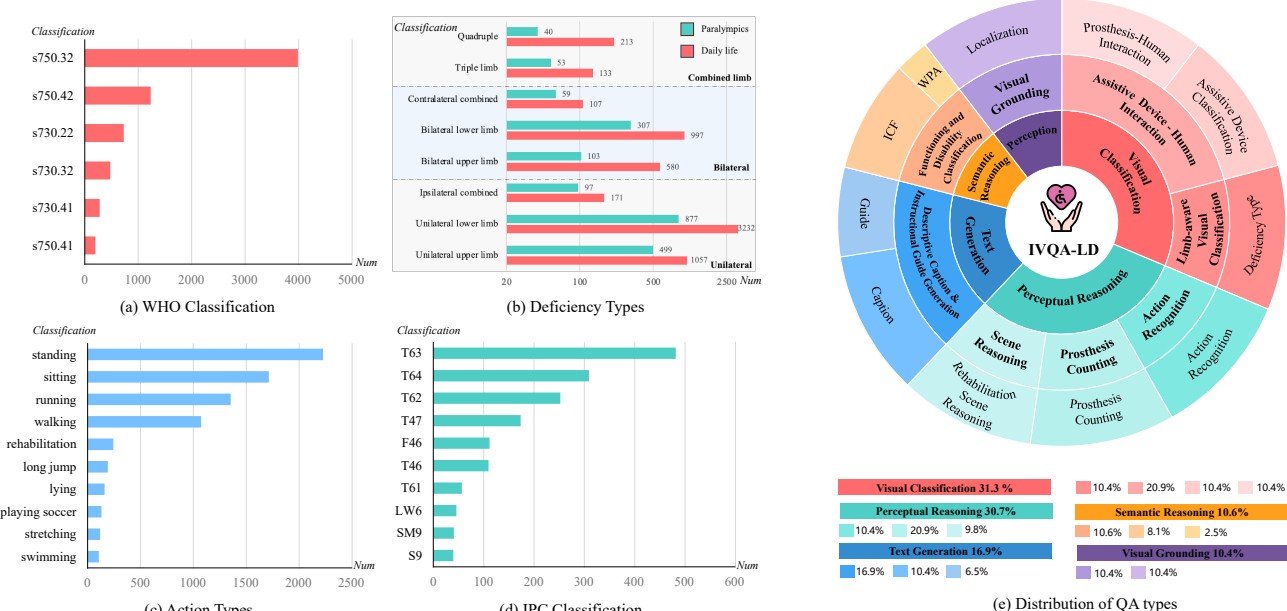

*Figure 2.* Statistics of IVQA-LD . (a) Distribution of WHO functional classes; (b) deficiency types; (c) action types; (d) Paralympic IPC classes; (e) distribution of QA types in our VQA benchmark.

task category to preserve the relative distribution of question types across all the subsets, ensuring a fair and consistent evaluation protocol. For captioning and rehabilitation guidance tasks, we ensure that all Q-A pairs in the test set are either authored or reviewed by domain experts, while machine-generated Q-A pairs are used exclusively for training, as expert annotation for these tasks is particularly time-consuming and costly.

### 3.5. Body-centric Structure-Aware Initialization Strategy for Fine-tuning

Inspired by how humans gradually acquire domain expertise when reasoning about limb deficiency, our proposed *Body-centric Structure-aware Initialization (BSI) strategy* explicitly incorporates limb- and body-structural priors through a two-stage fine-tuning process.

*Stage 1: Body-centric structural representation alignment.* We first warm up VLMs using limb-related appearance cues that only require general limb-deficiency knowledge, such as bounding boxes of residual limbs and spatial locations of prosthetic or assistive devices. In this stage, we freeze the vision encoder and the LLM backbone, only updating the parameters of the multimodal projector $\theta_p$:

$$\mathcal{L}_{S_1} = \mathcal{L}_{loc}(\mathbf{I}; \theta_p) = -\log P(\mathbf{B}|\mathbf{I}; \theta_p), \quad (1)$$

where $\mathbf{I}$ represents the input image, $\mathbf{B}$ denotes the ground-truth coordinates of the residual limbs or assistive devices, and $\mathcal{L}_{loc}$ denotes the localization loss. Here, $\mathcal{L}_{loc}$ employs cross-entropy loss to ensure compatibility with the token

prediction nature of MLLMs. This initialization enables VLMs to acquire sufficient knowledge on residual limbs, prosthetics and assistive devices.

*Stage 2: Grounding-enhanced fine-tuning.* We fine-tune the models on IVQA-LD by jointly supervising limb-related localization and disability-specific Q-A pairs, rather than relying solely on Q-A supervision. The model is required to generate the output in a structured format:

<bbox>$\hat{\mathbf{B}}$</bbox><answer>Text</answer>,

where $\hat{\mathbf{B}}$ represents the predicted bounding-box coordinates in the form of [xmin, ymin, xmax, ymax], corresponding to the top-left and bottom-right corners of the target region. When multiple residual limbs are present in an image, their bounding boxes are concatenated in the output sequence and predicted sequentially. In this stage, we update the joint parameters $\theta = \{\theta_p, \theta_{LLM}\}$ using a unified objective function:

$$\mathcal{L}_{S_2} = \mathcal{L}_{loc}(\mathbf{I}; \theta) + \sum_{t=1}^{T} \underbrace{-\log P(a_t|\mathbf{I}, \mathbf{Q}, \hat{\mathbf{B}}, a_{<t}; \theta)}_{\mathcal{L}_{vqa}}, \quad (2)$$

where $\mathcal{L}_{vqa}$ represents the next-token prediction loss for the answer sequence $\{a_1, ..., a_T\}$ conditioned on the predicted coordinates $\hat{\mathbf{B}}$ and the input question $\mathbf{Q}$. Our proposed training strategy encourages VLMs to attend to limb-deficiency-relevant regions before generating answers, and mitigates language-only shortcut learning as well as superficial ques-

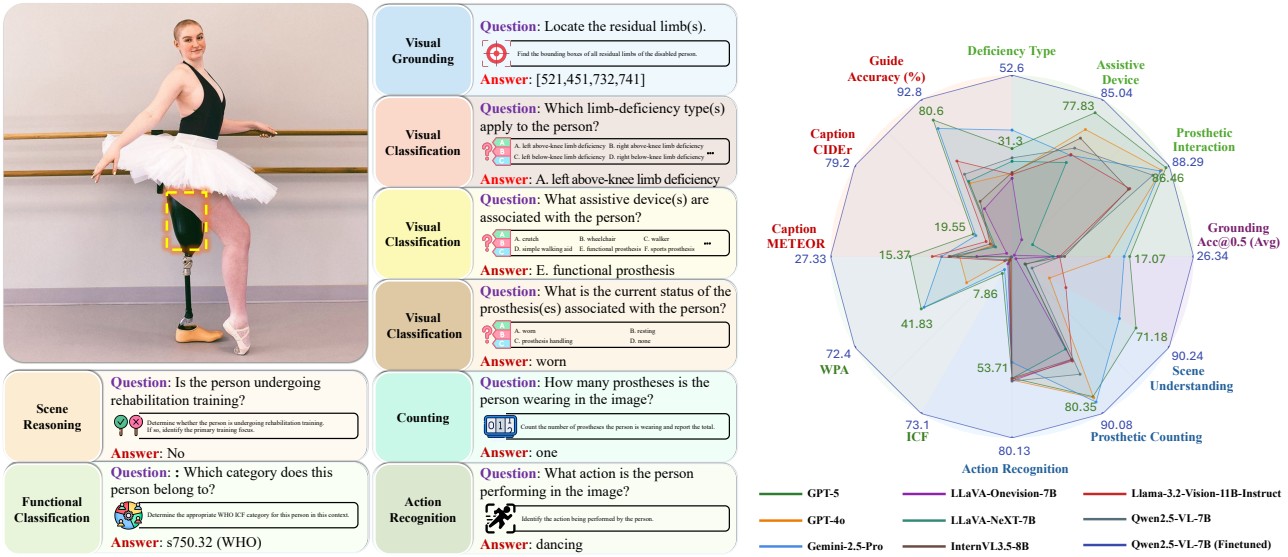

*Figure 3.* Illustration of representative IVQA-LD tasks with example inputs and outputs, together with model-wise performance comparison across limb-aware multimodal tasks.

tion–answer correlations, leading to significant performance improvements across various limb-deficiency tasks.

## 4. Experiments

### 4.1. Benchmark Tasks

IVQA-LD benchmarks eight tasks from four levels: **(i)** *Perception*: (1) visual grounding (VG); **(ii)** *Classification*: (2) deficiency type classification (DT), which serves as a limb-aware visual classification task, and (3) assistive device and prosthesis–human interaction understanding, which covers assistive device recognition (AD) and prosthesis–human interaction recognition (PI); **(iii)** *Reasoning*: (4) prosthesis counting (PC), (5) action recognition (AR), (6) scene reasoning (SR), and (7) functioning and disability classification (FDC); and **(iv)** *Generation*: (8) descriptive caption and instructional guide generation, which covers descriptive caption generation (DCG) and rehabilitation-specific instructional guide generation (IGG).

### 4.2. Evaluation Metrics

We employ task-specific evaluation metrics tailored to the different components of IVQA-LD, reflecting the distinct spatial, semantic, and generative requirements of limb-aware multimodal understanding. For localization-related tasks, we assess spatial accuracy using the **mean Intersection-over-Union (mIoU)** (Vakili et al., 2025) between predicted and ground-truth limb-deficiency regions. During evaluation, we extract all predicted boxes from the generated sequence and compute the mIoU against the ground-truth box set. For attribute-centric and classification-

based questions, we report **Top-1 Accuracy** over categorical labels, including limb-deficiency type, laterality, anatomical level, and functional or disability classes. For descriptive caption generation, we adopt standard vision-language evaluation metrics, including **CIDEr** (Vedantam et al., 2015) and **METEOR** (Banerjee & Lavie, 2005), which measure semantic relevance and linguistic consistency with expert-authored reference captions. Rehabilitation guidance generation is evaluated using an **LLM-as-a-Judge** framework (Gu et al., 2024; Li et al., 2025). Following patient-education evaluation principles (Shoemaker et al., 2014), each generated guide is assessed along seven dimensions: accuracy, appropriateness to the described condition, understandability, organization, actionability, supportive content, and caring tone, enabling fine-grained assessment of both clinical correctness and communicative quality.

### 4.3. Implementation Details

Our experiments are conducted using Qwen2.5-VL-7B-Instruct (Team, 2025) on eight NVIDIA A100-80GB GPUs, utilizing the LLaMA-Factory (Zheng et al., 2024) framework for efficient fine-tuning. For our proposed BSI strategy: we first freeze the vision tower and language model, updating only the multimodal projector $\theta_p$ via LoRA (Hu et al., 2021) ($r = 16$) for 1 epoch with a learning rate of $10^{-4}$ to establish spatial-structural priors. Then, we perform full-parameter fine-tuning for 5 epochs with a learning rate of $10^{-5}$. To ensure a fair comparison, all open-source baselines are evaluated using the LLaMA-Factory framework with a unified cutoff length of 5,120 tokens, while closed-source models are accessed via their respective official APIs. For generative tasks, we set the temperature to 0.7 and top-p to

*Table 2.* Comprehensive evaluation of model capabilities across multiple vision-language tasks. **VC**: Visual Classification (DT = Deficiency Type, AD = Assistive Device, PI = Prosthesis-Human Interaction); **PR**: Perceptual Reasoning (SR = Scene Reasoning, PC = Prosthesis Counting, AR = Action Recognition); **VG**: Visual Grounding (Acc@0.5); **FDC**: Functioning and Disability Classification (ICF = International Classification of Functioning, WPA = World Para Athletics Classification); **DCG**: Descriptive Caption Generation (M = METEOR, C = CIDEr); and **IGG**: Instructional Guide Generation (GS = Guide Score).

| | Classification | | | Perception | Reasoning | | | | | Generation | | |
|---|---|---|---|---|---|---|---|---|---|---|---|---|
| **Model** | **VC** | | | **VG** | **PR** | | | **FDC** | | **DCG** | | **IGG** |
| | DT | AD | PI | Acc@0.5 | PC | AR | SR | ICF | WPA | M | C | GS |
| *Closed-source Models* | | | | | | | | | | | | |
| GPT-5 | 31.30 | 77.83 | 86.46 | 17.07 | 71.18 | 80.35 | 53.71 | 7.86 | 41.83 | 15.37 | 19.55 | 4.03 |
| GPT-4o | 24.12 | 68.76 | 84.00 | 14.09 | 21.44 | 81.09 | 54.59 | 0.70 | 20.92 | 7.91 | 9.87 | 2.16 |
| Gemini-2.5-Pro | 36.66 | 54.77 | 83.28 | 16.27 | 61.69 | 83.43 | 46.75 | 6.07 | 40.52 | 10.60 | 18.21 | 3.78 |
| *Open-source Models* | | | | | | | | | | | | |
| LLaVA-Onevision-7B | 22.67 | 9.07 | 12.67 | 6.64 | 2.20 | 59.33 | 54.93 | 3.40 | 0.40 | 5.30 | 2.71 | 1.41 |
| LLaVA-NeXT-7B | 27.60 | 50.87 | 11.53 | 5.97 | 7.87 | 53.13 | 53.80 | 1.80 | 0.20 | 9.29 | 8.39 | 2.22 |
| Llama-3.2-Vision-11B-Instruct | 23.80 | 55.20 | 66.13 | 6.91 | 30.93 | 59.93 | 54.20 | 1.60 | 3.40 | 12.01 | 13.18 | 2.82 |
| InternVL3.5-8B | 24.33 | 64.07 | 65.40 | 7.62 | 7.40 | 59.00 | 53.73 | 2.50 | 0.40 | 9.48 | 11.00 | 1.63 |
| Qwen2.5-VL-7B | 28.60 | 58.67 | 79.07 | 7.18 | 11.47 | 67.53 | 55.07 | 0.30 | 2.80 | 8.17 | 9.22 | 2.43 |
| *Finetuned Models* | | | | | | | | | | | | |
| Qwen2.5-VL-7B | 37.60 | 80.13 | 67.33 | 18.91 | 86.67 | 71.60 | 83.00 | 73.11 | 63.80 | 26.04 | 65.37 | 4.54 |
| **Qwen2.5-VL-7B with *BSI*** | **52.60** | **85.04** | **88.29** | **26.34** | **90.24** | **90.08** | **80.13** | **75.40** | **72.40** | **27.33** | **79.20** | **4.64** |

*Table 3.* Ablation study on different training schemes. $S_1$ and $S_2$ denotes employing residual-limb and prothesis grounding supervision at stage 1 and 2, respectively.

| Training Scheme | DCG | | IGC |
|---|---|---|---|
| | METEOR | CIDEr | Score |
| SFT (w/o $S_1$ and $S_2$) | 26.04 | 65.37 | 4.54 |
| LoRA + SFT (w/ $S_1$) | 27.29 | 76.15 | **4.64** |
| LoRA + SFT (w/ $S_1$ and $S_2$) | **27.33** | **79.20** | **4.64** |

0.9 across all the models to maintain decoding consistency.

### 4.4. Baseline Models

**Closed-source models.** We evaluate three state-of-the-art proprietary multimodal systems: *GPT-5* (OpenAI, 2025), *GPT-4o* (Hurst et al., 2024), and *Gemini-2.5-Pro* (Comanici et al., 2025). These models represent the frontier in commercial multimodal reasoning.

**Open-source models.** We include several recent state-of-the-art open-source VLMs: *LLaVA-Onevision-7B* (An et al., 2025), *LLaVA-NeXT-7B* (Zhang et al., 2025), and *Llama-3.2-Vision-11B-Instruct* (Azaiz et al., 2025), *InternVL3.5-8B* (Wang et al., 2025), and *Qwen2.5-VL-7B-Instruct* (Team, 2025). These open-source models serve as transparent and reproducible baselines.

### 4.5. Main Results

We conduct an extensive evaluation of both closed-source and open-source MLLMs on the IVQA-LD. As shown in Table 2 and Figure 3(right), closed-source models in general outperform open-source ones in most cases, yet still exhibit notable failures specific to limb-deficiency perception. This phenomenon highlights that general-purpose multimodal models, even trained with massive scale of data, do not inherently endow themselves with the functional semantics for limb-deficiency populations.

Specifically, different models perform visual grounding of residual limbs, where the task requires detecting all residual-limb regions using bounding-box predictions and is measured by mIoU and ACC@0.5. Closed-source models perform much better than existing open-source systems, and GPT-5 achieves the best results. Open-source models generally performs poorly (under 8%), suggesting they struggle to localize residual-limb cues. Fine-tuning Qwen2.5-VL-7B-Instruct remarkably improves the accuracy to 26.34, even surpassing GPT-5. These results indicate that structured body- and limb-specific priors are essential for accurate residual-limb localization, enabling models to capture residual-limb-related visual evidence whereas general-purpose models usually fail to encode.

The disability classification contains two scenarios. For daily-life scenarios (following ICF protocols), closed-source models still have limited ability to align visual evidence with clinically grounded functional semantics, while for Paralympic scenarios (following WPA protocols) GPT-5 reaches better accuracy of 41.83% due to strong correlations with visible sports-context patterns. Open-source baselines remain near floor on both tasks, reflecting a pronounced domain gap. In contrast, fine-tuning model yields a dramatic improvement (73.10 on ICF and 72.40 on WPA), indicating that functioning and disability classification is a highly challenging semantic reasoning task for current VLMs. Furthermore, closed-source models achieve relatively strong performance in perceptual reasoning, especially for prosthetic-

status recognition, yet they continue to show weaknesses in functional semantics, *e.g.*, limb-affected action recognition. Open-source models perform much worse across all the perception tasks, suggesting that these models might not even access to sufficient limb-deficiency data.

Fine-tuning Qwen2.5-VL-7B-Instruct on IVQA-LD produces a marked improvement across all the aforementioned tasks. The model becomes more reliable in identifying limb-deficiency patterns, distinguishing assistive devices, and interpreting the operational state of a prosthesis, such as whether it is being worn, adjusted, or used for a specific action. These improvements also indicate that the model begins to internalize disability-specific semantics that are rarely encountered in general-purpose datasets. Furthermore, fine-tuning enhances the model's ability in WPA and ICF classification. These gains demonstrate that IVQA-LD provides the valuable domain-specific knowledge.

Closed-source models such as GPT-5 and Gemini-2.5-Pro achieve comparatively strong zero-shot captioning (*e.g.*, GPT-5 reaches 19.55 on CIDEr) and guide scores. Noted that GPT-4o performs worse in both captioning and assistive guidance, indicating that it fails to generalize to disability-grounded reasoning and suffers hallucination. Open-source models exhibit even a larger gap. For instance, LLaVA-OneVision-7B produces very low CIDEr and guide score, demonstrating difficulty in capturing disability-centric visual cues. Qwen2.5-VL and InternVL3.5-8B show moderate improvements but still fall far below closed-source systems.

### 4.6. Ablation Study

As shown in Table 3, we compare three training schemes to quantify the impact of our BSI strategy on both descriptive caption generation and instructional text generation tasks. Ablation results for the remaining tasks are reported in the supplementary material due to space limits. Our fine-tuning approach with BSI (LoRA+SFT w/ $S_1+S_2$) achieves the best performance. The results demonstrate that incorporating limb-deficiency-aware supervision in both stages enables MLLMs to better attend to limb-deficiency-relevant regions, leading to significant performance improvements on downstream tasks.

## 5. Conclusion

In this work, we introduced IVQA-LD, the first large-scale expert-annotated visual question answering benchmark designed to support limb-deficiency-aware multimodal learning and evaluation. IVQA-LD addresses a critical gap in existing vision-language resources by providing body-centric annotations, clinically grounded semantics, and diverse real-world scenarios covering daily life, rehabilitation, and Paralympic contexts. Through systematic benchmarking, we

demonstrated that state-of-the-art VLMs exhibit substantial limitations in limb-aware perception, reasoning, and generation, revealing persistent failure patterns in prosthesis understanding, residual-limb localization, and functional interpretation. Our proposed Body-centric Structure-aware Initialization strategy (BSI) that incorporates limb- and body-structural priors can effectively assist model adaptation in the limb-deficiency contexts, highlighting the importance of structured domain knowledge when adapting general-purpose multimodal models to disability-centered applications. We hope that IVQA-LD will serve as a foundational benchmark for future research on inclusive AI research.

## Acknowledgements

This research is funded in part by ARC-Discovery grant (DP220100800 to XY) and ARC-DECRA grant (DE230100477 to XY). We thank all anonymous reviewers and AC for their constructive suggestions.

## Impact Statement

This work aims to promote inclusive multimodal AI by introducing IVQA-LD, a limb-deficiency-aware VQA benchmark with expert-reviewed annotations. IVQA-LD may help reveal systematic limitations of current VLMs and support future models with better limb-aware multimodal understanding. However, models trained or evaluated on IVQA-LD may still produce inaccurate or insensitive outputs, especially in rehabilitation-oriented generation tasks. Therefore, this benchmark is intended for research and evaluation rather than clinical decision-making or unsupervised assistive deployment, and future use should involve human oversight and domain-expert validation.

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

This appendix is organized as follows:

- Broader Impact (Section A).

- Task Details (Section B).

- Experiment Details (Section C).

# A. Broader Impact

This work aims to improve inclusivity and fairness in multimodal AI systems by introducing a dataset and benchmark focusing on individuals with limb deficiency, a population that is significantly underrepresented in existing vision-language resources. By providing body-centric annotations and clinically grounded semantic supervision, our dataset may support the development of assistive AI technologies, improve robustness of multimodal models in real-world healthcare and rehabilitation-related scenarios, and help reduce systematic performance disparities across different population groups (Sheng et al., 2024; 2026; Shen et al., 2025a). More broadly, this work contributes to ongoing efforts toward responsible and representative AI by highlighting the importance of domain-specific data for underrepresented communities.

At the same time, we acknowledge several potential risks. First, models trained on this dataset are not intended for clinical decision-making, and misuse in medical or rehabilitation contexts may lead to incorrect or unsafe recommendations. Second, despite careful data curation and quality control, residual biases may still exist due to limitations in data coverage, demographic diversity, and real-world variability. Third, multimodal models may generate plausible but incorrect medical or functional interpretations, which could be harmful if interpreted as expert guidance. To mitigate these risks, our dataset is released for research purposes, and we emphasize that downstream systems should incorporate human oversight and domain expert validation when deployed in sensitive applications. We hope this work encourages further research on safe, fair, and inclusive multimodal AI systems.

# B. Task Details

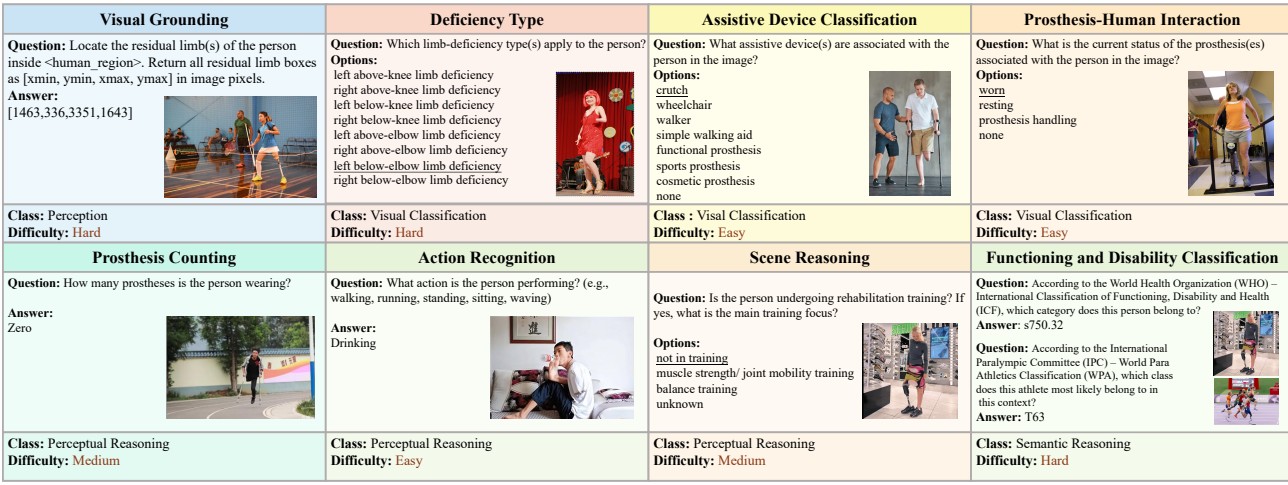

*Figure 4.* Examples of VQA Tasks in the Dataset

The Visual Question Answering (VQA) tasks in this study are organized into a four different hierarchy, as shown in Fig. 4, ranging from perception to generation. This structure, detailed below, systematically reflects the increasing cognitive complexity and reliance on external domain knowledge required to accurately solve each task.

**Perception Class.** For this task category, we specifically formulate the challenge as a visual grounding problem. This is a hard task, focusing on region recognition. It verifies the model's ability to recognize *what* an object is and *where* it is located in the image space. The challenge lies in dealing with non-standard objects (residual limbs) and partial occlusions common in real settings, but the required output is spatial (coordinates). This task is critical, as it provides the spatial foundation for all subsequent analyses.

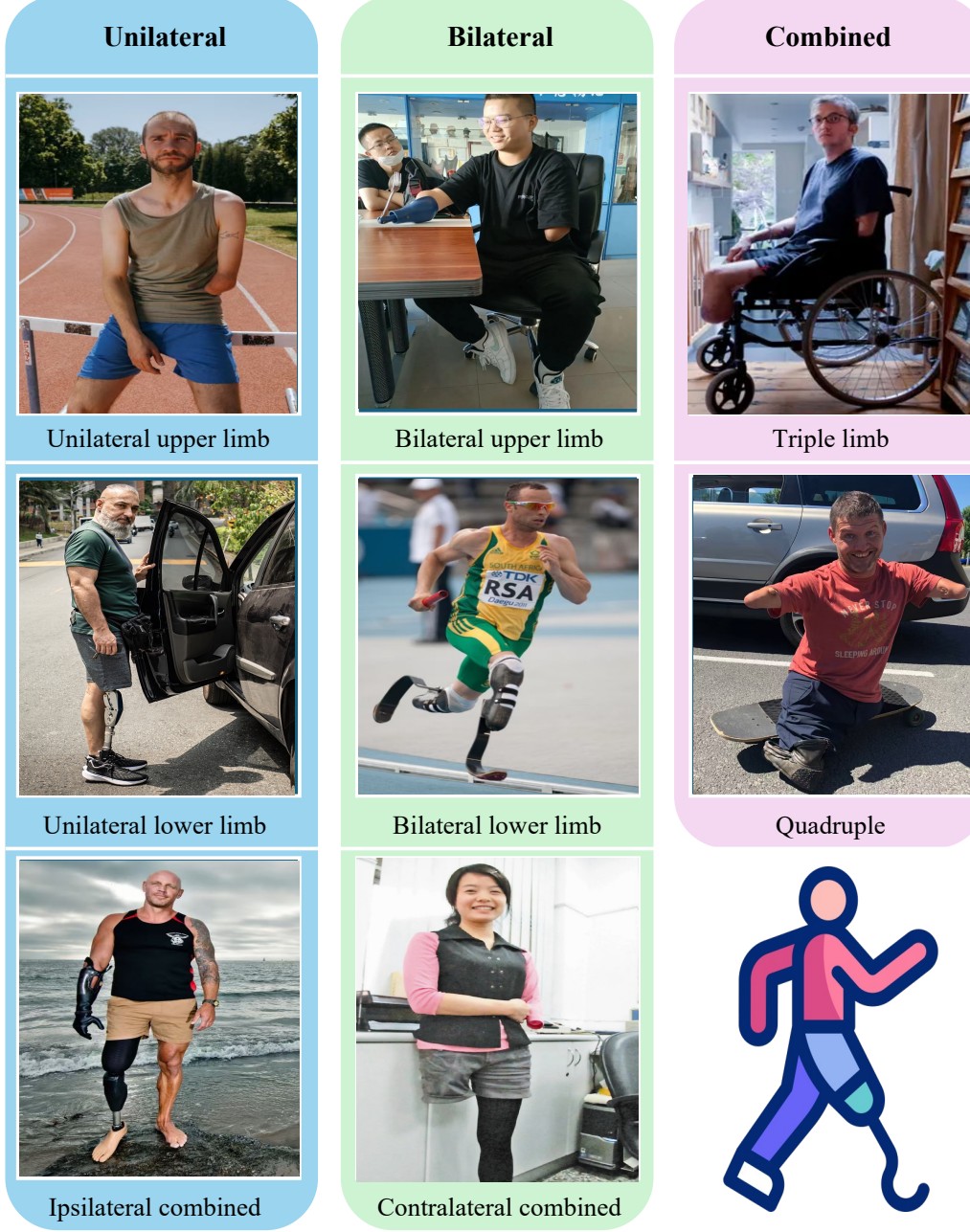

*Figure 5.* Examples of All Deficiency Types

**Classification Class.** This category comprises two problem types: deficiency type classification and assistive device classification. Assistive device classification is treated as an easy-level task, as it mainly relies on recognizing visible assistive objects, such as wheelchairs, crutches, or prosthetic devices. In contrast, deficiency type classification is more challenging because it requires limb-aware understanding of fine-grained limb-deficiency configurations, including laterality, affected limb type, and anatomical level, such as left/right, upper/lower limb, and above-/below-knee or elbow distinctions. These judgments can be affected by occlusion, viewpoint variation, clothing, prosthesis use, and partial body visibility. Accuracy therefore depends not only on recognizing visible objects, but also on extracting subtle visual cues and distinguishing between closely related deficiency categories. As shown in Fig. 5, limb-deficiency configurations are organized according to laterality, affected limb type, and anatomical level. Accordingly, our QA design uses fine-grained options that specify concrete visible configurations, such as left/right involvement and above-/below-knee or above-/below-elbow deficiency, rather than

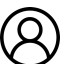

Write a caption describing the person with limb loss shown in the image.

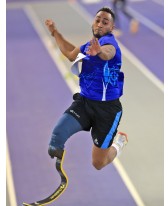

InternVL3.5

An athlete in a blue and black uniform is captured mid-air during a jump. He has a prosthetic leg and is wearing white shoes, showcasing determination and athleticism on the track.

**Ground Truth**

A para-athlete wearing a red-and-blue athletic outfit performs a long jump during an athletics event, captured mid-air above the runway. The para-athlete competes in the F64 classification and uses a running blade sports prosthesis on one leg while wearing a blue top and black shorts with arms extended for balance.

*Figure 6.* Visualization of Model Performance on the Caption Generation Task.

GPT-4o

......Practice moving safely with the help of nurses or therapists, and remember that play can be part of therapy. ...... Tell the nurses if something hurts so they can try to help you feel better. ...... Keep bandages clean and dry, and tell a grown-up if you notice redness, swelling, or a bad smell. ......

Write a rehabilitation guide for this person.

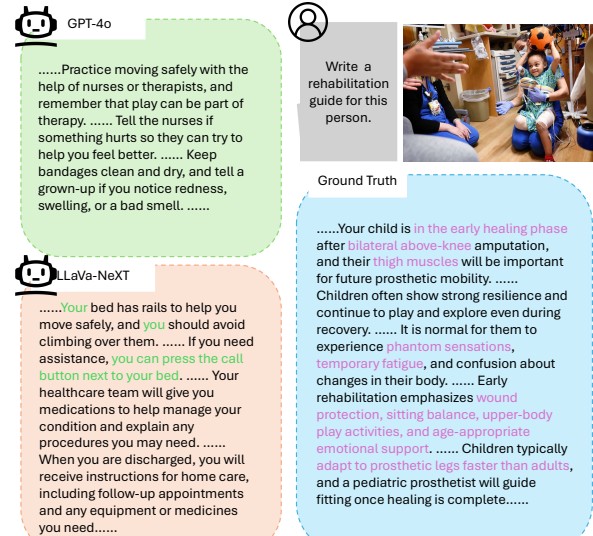

**Ground Truth**

......Your child is in the early healing phase after bilateral above-knee amputation, and their thigh muscles will be important for future prosthetic mobility. ...... Children often show strong resilience and continue to play and explore even during recovery. ...... It is normal for them to experience phantom sensations, temporary fatigue, and confusion about changes in their body. ...... Early rehabilitation emphasizes wound protection, sitting balance, upper-body play activities, and age-appropriate emotional support. ...... Children typically adapt to prosthetic legs faster than adults, and a pediatric prosthetist will guide fitting once healing is complete......

LLaVa-NeXT

......Your bed has rails to help you move safely, and you should avoid climbing over them. ...... If you need assistance, you can press the call button next to your bed. ...... Your healthcare team will give you medications to help manage your condition and explain any procedures you may need. ...... When you are discharged, you will receive instructions for home care, including follow-up appointments and any equipment or medicines you need......

*Figure 7.* Visualization of Model Performance on the Rehabilitation Guide Generation Task.

relying only on coarse labels such as unilateral or bilateral limb deficiency. Such distinctions support rehabilitation-aware interpretation of visual scenes and functional evaluation of assistive AI systems.

**Reasoning Class.** This category includes prosthesis counting, action recognition, scene reasoning, and functioning and disability classification. These tasks go beyond direct visual classification and require models to integrate visual evidence with relational and domain-aware reasoning. For example, prosthesis counting and action recognition require localizing relevant body parts, prosthetic devices, and actions, while scene reasoning requires interpreting rehabilitation-related activities and contexts. Functioning and disability classification is more domain-specific, as it asks the model to relate observable functional cues to established functioning-oriented classification schemes. Overall, this class evaluates whether models can reason over limb-deficiency-related visual evidence rather than simply recognize isolated objects or categories.

**Generation Class.** This category includes descriptive caption generation and rehabilitation-specific instructional guide generation. These tasks require open-ended responses that are accurate, coherent, respectful, and grounded in the visual content. Descriptive caption generation focuses on summarizing the visible person, action, limb-deficiency or prosthesis-related cues, assistive devices, scene context, and other relevant visual details. Instructional guide generation further requires rehabilitation-aware guidance that is grounded in the visual evidence and supported by relevant domain knowledge. This class evaluates the model's ability to generate informative, respectful, and responsible language for limb-deficiency-related scenarios. The prompts used for generating the captions and guides in this training data are shown in Fig. 8 and Fig. 9.

Fig. 6 and Fig. 7 further illustrate the qualitative behavior of models on the two generation tasks. In the caption generation example, the model captures the general athletic action but misses or misstates key prosthesis-related details, showing that accurate captions require more than generic scene description. In the instructional guide generation example, GPT-4o provides plausible but generic rehabilitation advice, with limited grounding in the child's visible condition, rehabilitation stage, and caregiver-oriented context. LLaVA-NeXT (Liu et al., 2024b) further produces image-ungrounded advice by mentioning a bed and a call button that are not visible, and it addresses the guide as if it were for an adult patient rather than a caregiver of a young child. In contrast, the ground truth is more closely grounded in the visible condition, rehabilitation stage, and child-specific care context. These examples show that generation tasks require both visual faithfulness and domain-aware, respectful language.

## C. Experiment Details

### C.1. Training Details

To ensure robust multimodal alignment, all experiments were conducted using the Qwen2.5-VL-7B-Instruct architecture on an AMD EPYC 9224 (64 cores) system equipped with eight NVIDIA A100-SXM4-80GB GPUs, 1 TB RAM, and NVMe

*Table 4.* Evaluation metrics and judging methods for different question types in ABLE-VQA.

| Question Type | Evaluation Metric | Judging Method |
|---|---|---|
| Visual Grounding | Acc@0.5 | Automatic |
| Multiple Selection | Exact-match | LLM + Regex |
| Counting | Numeral/word-form match | Regex |
| Open-ended | Semantic consistency | LLM |
| Caption | CIDEr and METEOR | Automatic |
| Guide | Expert 1–5 score | LLM |

SSD storage. The DeepSpeed ZeRO-3 framework was employed to partition parameters, gradients, and optimizer states across GPUs, enabling memory-intensive full-parameter optimization. The Baseline protocol adopted standard Supervised Fine-Tuning (SFT) on VQA pairs without explicit spatial grounding constraints ($lr = 1\mathrm{e}{-5}$, 5 epochs, batch size = 8), serving as a spatial-agnostic control.

Our proposed Methods 1 and 2 followed a Two-Stage Fine-Tuning Curriculum that disentangles structural localization from semantic reasoning. Stage 1 applied LoRA ($r = 16$, $target = all$, $lr = 1\mathrm{e}{-4}$) solely on bounding-box supervision to establish a robust spatial prior over residual-limb regions. Stage 2 then performed full-parameter SFT ($lr = 1\mathrm{e}{-5}$, 5 epochs) on the complete multimodal corpus. Method 2 uniquely introduced an interleaved spatial-reasoning mechanism during Stage 2, conditioning the model to regress limb coordinates before each response generation; this hierarchical attend-and-reason pattern enforces persistent visual grounding and substantially reduces hallucination in extended diagnostic dialogues.

## C.2. LMMs for Evaluation

In this paper, we evaluate the perception, reasoning and generation capabilities of the following VLMs on the IVQA-LD benchmark:

- **LLaVA-OneVision-7B.** An open-source multimodal model based on the Qwen2 language model. It is designed to handle single-image, multi-image, and video understanding tasks.
- **LLaVA-NeXT-7B.** An open-source multimodal model from the LLaVA family, designed to improve visual reasoning, OCR, and instruction following. In our experiments, we use the 7B variant.
- **Llama-3.2-Vision-11B.** An open-weight vision-language model from Meta. It is optimized for visual recognition, image reasoning, captioning, and general visual question answering.
- **InternVL3.5-8B.** An open-source multimodal model developed by OpenGVLab. It is designed to improve multimodal reasoning, versatility, and inference efficiency.
- **Qwen2.5-VL-7B.** An open-source vision-language model from the Qwen series. It is designed for visual understanding tasks such as document parsing, object recognition, and visual question answering.
- **GPT-4o.** A closed-source multimodal model developed by OpenAI. It can reason across text, vision, and audio inputs, and is widely used as a strong proprietary baseline.
- **GPT-5.** A closed-source frontier multimodal model developed by OpenAI. It shows strong capabilities in multimodal reasoning, including visual, video-based, spatial, and scientific reasoning.
- **Gemini 2.5 Pro.** A closed-source multimodal model developed by Google DeepMind. It is designed for complex reasoning, coding, and multimodal understanding tasks.

## C.3. Evaluation Metrics

For each question category, we employed a tailored evaluation metric based on its answer format and content, as shown in Table 4. Visual grounding responses were scored via bounding-box Intersection over Union (IoU): a prediction was deemed correct if its IoU with the ground-truth box exceeded 0.5 (reported as Acc@0.5). For multiple-select QA, GPT-4.1-mini is first used for semantic option alignment, and regex-based exact matching is then applied to compare the predicted option set with the gold set. Counting questions were evaluated using regular expressions to match numeric equivalence, accepting both Arabic numerals and English word forms (e.g., "0" or "zero," "5" or "five"). Open-ended textual answers were judged by GPT-4.1-mini, which compared the predicted answer with the reference answer for semantic equivalence and returned a binary correctness judgment. Image captioning performance was measured using standard caption quality metrics:

**Caption Generation Prompt**

**Task:** You are given: 1) A cropped image and a full image focusing on the target individual; 2) Structured metadata for the target individual. Write ONE paragraph as an objective observation, similar in tone and structure to the Example Reference below.

 **Rules:**
- Describe what is visually observable and avoid speculation.
- The metadata can be used to improve precision.
- No recommendations or treatment plans.

**Examples:**
<In context learning>

**Metadata:**
<Ground truth data>

**Output requirements:**
- One paragraph.
- Factual, neutral tone.
- No lists, no headers.

-------------
Now generate the caption for this person.

*Figure 8.* Caption Generation Prompt

CIDEr and METEOR scores were computed for each generated caption against the reference using the COCO caption evaluation toolkit. Finally, for guide-type responses, GPT-4.1-mini was prompted with a specialized rubric to compare model predictions to gold-standard guides and assign a correctness score from 1 to 5, as shown in Fig. 10.

**Guide Generation Prompt**

**Task:** Based on the image and the following caption, write one Rehabilitation Guide that helps the person and their family understand the condition and what they can do. Reference materials and examples are provided for you to use.

**Rules:**
- Use plain, compassionate English that can be easily understood by patients and families.
- Keep the tone supportive, positive, and empowering.
- Avoid medical terminology

**Examples:**
<In context learning>

**Metadata:**
<Caption>

**Output requirements:**
- Follow the structure and formatting style shown in the example above.

-------------

Now generate the  guide for this person.

*Figure 9.* Guide Generation Prompt

**Guide Evaluation Prompt**

**Task:** You are an expert evaluator of medical rehabilitation guides. You will receive a 'Gold Answer' (which represents the 5-point standard) and a 'Predicted Answer'. Your task is to compare the 'Predicted Answer' with the 'Gold Answer' and score each of these 7 criteria from 1 to 5, along with an average score:
1.Factual Accuracy (accuracy): Is the prediction as medically accurate as the gold standard?
2.Appropriateness (appropriateness): Is the prediction as appropriate for the described condition as the gold standard?
3.Understandability (understandability): Is the prediction as clear and easy to understand as the gold standard?
4.Organization (organization): Is the prediction as well-organized as the gold standard?
5.Actionability (actionability): Does the prediction provide actionable guidance like the gold standard?
6.Caring Tone (caring tone): Does the prediction have a caring and compassionate tone like the gold standard?
7.Supportive Content (supportive content): Does the prediction provide supportive content like the gold standard?

**Rules:**
Scoring Rubric:
• 5 (Excellent): The prediction fully meets the standard of the Gold Answer.
• 4 (Good): The prediction largely meets the standard but is slightly inferior to the Gold Answer.
• 3 (Fair): The prediction is of moderate quality. Understandable, but lacks the detail, precision, or relevance of the Gold Answer.
• 2 (Poor): The prediction contains unclear or partially inaccurate information, with limited relevance.
• 1 (Very Poor): The prediction is factually wrong, misleading, or inappropriate.

**Output requirements:**
Return only the final average score.

-------------

Start the evaluation now.

*Figure 10.* Guide Evaluation Prompt

