# OpenReview forum: "IVQA-LD: Inclusive Multimodal Understanding for Population with Limb Deficiency"
_ICML.cc/2026/Conference — ICML 2026 regular_

### Official Review · Reviewer_Cwyg · 2026-03-10

**Soundness:** 3
**Presentation:** 3
**Significance:** 3
**Originality:** 3
**Overall Recommendation:** 4
**Confidence:** 4

**Summary:**

This paper introduces IVQA-LD, a limb-deficiency-aware, body-centric multimodal benchmark for evaluating vision-language models on an underrepresented population. The dataset contains 14,054 curated real-world images and 80K VQA pairs spanning eight tasks, including visual grounding, classification, reasoning, captioning, and rehabilitation-oriented guide generation. The paper also proposes a Body-centric Structure-aware Initialization (BSI) strategy, which first warms up the model with limb- and device-localization cues and then performs grounding-enhanced fine-tuning for question answering. Experiments on both open- and closed-source VLMs show that existing models struggle in this setting, while Qwen2.5-VL-7B fine-tuned on IVQA-LD, especially with BSI, achieves substantial gains across multiple tasks.

**Compliance With Llm Reviewing Policy:**

Affirmed.

**Key Questions For Authors:**

Q1. Clarification on closed-source baselines and reproducibility:
The main paper reports GPT-5 as a benchmarked closed-source baseline, while Appendix C.2 describes it as a forthcoming/hypothetical model. Could the authors clarify what exact system/version was actually evaluated, and how this comparison can be made reproducible for  readers?

Q2. Details of the grounding/tokenization pipeline:
The paper states that residual-limb localization is learned through structured output and cross-entropy supervision, but key implementation details remain unclear. How are coordinates represented, what is the exact output grammar, and how are multiple residual limbs handled during both training and evaluation?

Q3. Clarification and validation of the evaluation protocol:
Could the authors clarify exactly which task categories rely on GPT-4.1-mini judging, since Appendix C.3 appears to describe both rule-based scoring and LLM-based evaluation for different non-grounding tasks? In addition, was the LLM judge validated against expert raters on a subset of examples, especially for rehabilitation-oriented guidance?

Q4. Split protocol and source-level leakage control:
The paper notes that the raw source contains many frames from the same videos and that near-duplicate filtering was applied. Were the train/validation/test splits grouped by source video, event, or individual identity? If not, could the authors quantify possible overlap and provide results under a stricter split protocol?

Q5. Ablation coverage across task categories:
The main paper reports ablations mainly for generation tasks, while the broader claim is that BSI improves performance across the full benchmark. Could the authors include at least a compact ablation table for representative understanding tasks such as visual grounding, functioning/disability classification, and action recognition?

**Limitations:**

YES

**Strengths And Weaknesses:**

Strengths
(S1) The paper addresses an important and underexplored inclusivity problem. Existing VLM benchmarks rarely cover people with limb differences, and IVQA-LD fills a clear gap by focusing on body-centric functional understanding and assistive-device reasoning.

(S2) The dataset design is thoughtful and practically meaningful. The task taxonomy spans perception, classification, reasoning, and generation, which makes the benchmark useful not only for performance ranking but also for diagnosing where current VLMs fail.

(S3) The annotation pipeline is reasonably well designed. The use of WHO ICF and Paralympic-related functional standards gives the benchmark stronger domain grounding than generic assistive VQA datasets, and the paper describes multi-stage quality control involving both non-expert auditing and expert review.

(S4) The proposed BSI strategy is simple, task-relevant, and empirically effective. The idea of learning “where to look” before “what to answer” is intuitive, and the reported gains suggest that body-centric structural supervision is useful in this setting.

Weaknesses
(W1) The evaluation protocol is not fully clear and appears internally inconsistent. In Appendix C.3, multiple-select questions are first described as using exact-match accuracy, while the later summary and Table 4 suggest that multiple-select, open-ended, and guide tasks rely on GPT-4.1-mini for evaluation. This ambiguity, together with the use of LLM-based judging for several tasks, makes the benchmark harder to interpret unless the authors clearly specify which tasks use rule-based versus LLM-based evaluation and provide stronger validation against human experts.
(W2) Important methodological details of the grounding pipeline are still underspecified. The paper states that bounding boxes are predicted in token form and supervised with cross-entropy, but it does not clearly explain coordinate discretization, output constraints, or how multiple residual limbs are matched during training and evaluation. These details are important for both reproducibility and for understanding why the method works.

(W3) There are presentation and consistency issues that should be cleaned up. In particular, the appendix refers to “ABLE-VQA” while the main paper uses “IVQA-LD,” and the use of WPA/IPC terminology is not always consistent. These issues are not fatal, but they do reduce clarity.

(W4) The split protocol and ablation evidence could be stronger. The paper notes that the source data include many images extracted from the same videos and that near-duplicate frames are filtered, but it is not clear whether train/val/test splits are grouped by source video or identity. In addition, the main-paper ablation only reports generation tasks, while the broader claim is that BSI improves performance across all task categories.

---

> ### Author Rebuttal · Authors · 2026-03-31
>
> We sincerely thank **Reviewer Cwyg** for the constructive comments and for recognizing the inclusivity value of IVQA-LD, our rigorous dataset design, and the empirical effectiveness of our BSI strategy. We address your specific concerns below.
> >**Q1:** Clarification on the versions of closed-source baselines and presentation issues.
>
> **R1:** Thank you for the suggestion.
> - **Closed-source Baselines**: The specific API versions of the closed-source models evaluated in our main experiments (Table 2) are as follows: gpt-4o-2024-08-06, gpt-5-2025-08-07, gemini-2.5-Pro (stable release, accessed in Nov. 2025).
> - **Terminology Consistency**: We will correct the typo (replacing 'ABLE-VQA' with 'IVQA-LD') and ensure consistency throughout the manuscript.
>
> We will include the exact model versions and correct typos in the revision.
>
> >**Q2:** Details of the grounding/tokenization pipeline.
>
> **R2:** To clarify the implementation and ensure reproducibility, our grounding pipeline follows the native spatial representation of our base model (Qwen2.5-VL):
>
> - **Coordinate Discretization**: Following Qwen2.5-VL's native architecture, we convert continuous spatial locations into a discrete format suitable for language modeling.
> - **Output Grammar & Constraints**: During the structured output phase, the model is constrained to generate coordinates in the format **\<box> [xmin, ymin, xmax, ymax] \</box>**.
> - **Handling Multiple Limbs**: When multiple residual limbs are present in a single image, the ground-truth sequence is constructed by concatenating the bounding boxes (*e.g.*, **\<box> [xmin_1, ymin_1, xmax_1, ymax_1] \</box> \<box> [xmin_2, ymin_2, xmax_2, ymax_2] \</box>**). The model is trained to sequentially predict all relevant regions. During evaluation, we extract all predicted boxes from the generated sequence and compute the mean Intersection-over-Union (mIoU) against the ground-truth set.
>
> We will incorporate such detailed implementations into the revised paper.
>
> >**Q3:** Clarification and validation of the evaluation protocol.
>
> **R3:** To clarify, our evaluation protocol is as follows:
> - **Automated/Rule-based**: Visual grounding (IoU / Acc@0.5), counting (numeric matching), and captioning (CIDEr / METEOR).
> - **LLM-based**: Used for three specific cases: (1) multiple-select QA, (2) open-ended semantic QA, and (3) rehabilitation-oriented guidance.
>
> Our benchmark accepts options as raw text (*e.g.*, "walker") rather than lettered labels (*e.g.*, "A. walker; B. wheelchair;..."). Therefore, we use GPT-4.1-mini strictly to align model responses with the corresponding semantic option text. The alignment enables a structured "semantic exact match" against the ground truth. By doing so, we not only determine whether a question has been answered correctly, but also identify the specific incorrect option when the response is wrong, thus providing more informative error analysis beyond simple accuracy.
>
> To assess the reliability of the LLM-based evaluation, five experts validate 500 samples in the Instructional Guide Generation (IGG) task, following the protocol in Fig. 8. Table A compares expert ratings with LLM scores. The strong Pearson correlation ($r$) demonstrates that our framework is a reliable proxy for expert assessment.
>
> **Table A**: Comparisons between human expert ratings and LLM-as-a-judge scores across different models, along with their Pearson correlation coefficients.
>
> |Model|Expert Score (Avg)|LLM-as-a-Judge Score (Avg)|Correlation (Pearson $r$)|
> |-|-|-|-|
> |GPT-5| 3.77|4.03|0.82|
> |Gemini-2.5-Pro|3.25|3.78|0.75|
> |Qwen2.5-VL-7B (Zero-Shot)|2.05|2.43|0.88|
> |Qwen2.5-VL-7B (SFT)|4.11|4.54|0.78|
> |Qwen2.5-VL-7B (BSI SFT)|4.28|4.64|0.84|
>
> >**Q4:** Split protocol and source-level leakage control.
>
> **R4:** Our splitting protocol is strictly conducted at the source **video** level instead of the individual frame level. The strategy prevents frames from the same video sequence from crossing over between subsets. We also manually discarded near-duplicate frames to eliminate potential overlap. We will state the video-level splitting protocol in the revised version.
>
> >**Q5:** Ablation coverage across task categories.
>
> **R5:** As suggested, we conduct additional ablation studies and analyse the impact of our two-stage BSI strategy on nine different tasks, as shown in Table B. The results indicate that body-centric structural initialization provides critical spatial anchoring that leads to improved performance on downstream tasks. We will include these results and analysis in the revised version.
>
> **Table B**: Ablation study of our BSI strategy across tasks defined in Table 2 of the manuscript.
>
> |Training Scheme|DT|AD|PI|VG|PC|AR|SR|ICF|WPA|
> |-|-|-|-|-|-|-|-|-|-|
> |SFT (w/o Stage 1)|37.60|80.13|67.33|18.91|86.67|71.6|83.00|73.11|63.80|
> |LoRA + SFT (w/ Stage 1 only)|42.00|81.93|68.27|19.27|88.13|72.87|79.27|74.20|64.60|
> |LoRA + SFT (w/ Stage 1 and Stage 2)|52.60|85.04|88.29|26.34|90.24|90.08|80.13|75.40|72.40|

---

> > ### Author Rebuttal · Reviewer_Cwyg · 2026-04-02
> >
> > The rebuttal has addressed my concerns and significantly improved the clarity for the paper’s claims. I will maintain my score.

---

### Official Review · Reviewer_cug1 · 2026-03-12

**Soundness:** 3
**Presentation:** 3
**Significance:** 3
**Originality:** 3
**Overall Recommendation:** 4
**Confidence:** 3

**Summary:**

This paper focuses on multimodal understanding for populations with limb deficiency. On the data side, it introduces the first large-scale limb-deficiency-aware multimodal VQA dataset, IVQA-LD. On the methodological side, the authors propose a Body-centric Structure-aware Initialization strategy that explicitly injects body-structural priors into the model. Experimental results show that, after fine-tuning on IVQA-LD, VLMs achieve substantial performance improvements across multiple tasks.

**Compliance With Llm Reviewing Policy:**

Affirmed.

**Final Justification:**

For Q1, I thank the authors for the detailed clarification and additional experimental results. The response clearly articulates the distinction between the proposed method and “thinking-with-images” approaches, and the provided comparisons help better highlight the advantages of the proposed strategy.

For Q2, I appreciate the authors’ efforts in clarifying the annotation process and providing additional analysis. While the explanation is helpful, I still believe that incorporating a larger proportion of expert annotations would further strengthen the reliability and overall quality of the dataset.

Overall, the rebuttal offers valuable additional insights and clarifications. Based on this, I have increased my initial score.

**Key Questions For Authors:**

Please refer to the weakness part.

**Limitations:**

Yes.

**Strengths And Weaknesses:**

[+] The limb-deficiency scenario addressed in this paper carries clear fairness and inclusivity value, highlighting important societal significance.

[+] IVQA-LD is the first multimodal VQA benchmark specifically targeting populations with limb deficiency. The dataset is of sufficient scale, and the annotation process is rigorously designed.

[-] Although the proposed Body-centric Structure-aware Initialization is well-motivated and reasonably designed, similar structural strategies have been applied in grounding-aware VLM adaptation and “thinking with images” approaches. The methodological contribution lies more in the application context than in algorithmic novelty.

[-] Approximately 90% of the QA pairs are generated by non-expert annotators. While quality control mechanisms are in place, it remains unclear whether this is sufficiently rigorous for complex semantic tasks. Moreover, performance comparisons between expert-authored and non-expert subsets are not provided.

---

> ### Author Rebuttal · Authors · 2026-03-31
>
> We sincerely thank **Reviewer cug1** for the constructive comments and for recognizing the clear fairness and inclusivity value, the sufficient scale, and the rigorous annotation process of our IVQA-LD benchmark. We address the reviewer's concerns below.
>
> >**Q1:** Differences from "thinking with images" and ground-aware VLM adaption approaches.
>
> **R1:** The key difference is that BSI is a **training-time** body-centric structural initialization that injects anatomically meaningful inductive bias into the multimodal alignment process, rather than an **inference-time** crop/zoom policy [1] or a generic grounding prompt [2]. More importantly, the core challenge in our benchmark is not merely to “locate a small region”, but to jointly reason about residual limb location, prosthesis-human interaction, and full-body action and function context. In such scenarios, generic zoom-in may remove global cues required for captioning, guide generation, and action understanding.
>
> As suggested, we compare our benchmark with a representative 'thinking with images' baseline, DeepEyes [3], in Table A. The results show that such generic approaches often yield limited gains or struggle with complex semantic tasks, as they may discard critical global context.  In contrast, our BSI strategy introduces anatomically grounded inductive bias, consistently improving performance across both localization and reasoning tasks.
>
> We agree with the Reviewer that our BSI strategy falls under grounding-aware VLM adaptation. However, such adaptation requires the clinical and domain-specific knowledge in our IVQA-LD dataset. In particular, our dataset enables learning anatomically meaningful and context-aware alignments not accessible from generic sources. Therefore, a key contribution lies in the curation of IVQA-LD and its role in enabling effective adaptation of VLMs to better support populations with limb deficiencies. We will include this in the revision.
>
> **Table A**: Ablation study of BSI strategy across multiple tasks, including Deficiency Type (DT), Assistive Device (AD), Prosthesis-Human Interaction (PI), Visual Grounding (VG), Prosthetic Counting (PC), Action Recognition (AR), Scene Reasoning (SR),  International Classification of Functioning (ICF), World Para Athletics Classification (WPA) and Generation (M=METEOR, C=CIDEr, GS=Guide Score).
>
> |Model|Strategy|DT|AD|PI|VG|PC|AR|SR|ICF|WPA|M|C|GS|
> |-|-|-|-|-|-|-|-|-|-|-|-|-|-|
> |Qwen2.5-VL-7B|Zero-Shot|28.60|58.67|79.07|7.18|11.47|67.53|55.07|0.30|2.80|8.17|9.22|2.43|
> |DeepEyes (Qwen2.5-VL-7B)|Thinking with Images|20.36|62.11|41.49|18.81|57.22|44.96|24.55|1.81|1.03|6.19|4.90|1.29|
> |Qwen2.5-VL-7B| BSI (Ours)|52.60|85.04|88.29|26.34|90.24|90.08|80.13|75.40|72.40|27.33|79.20|4.64|
>
> [1] P. Wu, et al., V*: Guided Visual Search as a Core Mechanism in Multimodal LLMs. CVPR 2024.
>
> [2] K. Chen, et al., Shikra: Unleashing Multimodal LLM's Referential Dialogue Magic. Arxiv 2023.
>
> [3] Z. Zheng, et al., DeepEyes: Incentivizing "Thinking with Images" via Reinforcement Learning. ICLR 2026.
>
> >**Q2:** Concerns about ~90% non-expert annotations for complex tasks and performance comparisons between expert and non-expert subsets.
>
> **R2:** We would like to clarify the ~90% non-expert annotations correspond to objective, low-level perceptual tasks that do not require clinical expertise (*e.g.*, drawing bounding boxes for residual limbs, identifying assistive devices, and counting prosthetics). Furthermore, all non-expert annotators have received prior training from biomechanics experts and rehabilitation professionals to ensure annotation consistency and quality. As described in Page 4 (L187–203, right column), we also implemented additional quality control procedures for non-expert annotations to further ensure their reliability.
>
> Please note that the visual perception, classification, and reasoning tasks (Levels 1–3, Page 5, L249–252) do not require clinical expertise. These tasks are objective and can be reliably performed by trained non-expert annotators. As suggested by Reviewer **cug1**, we additionally ask experts to independently annotate around 5% of the data. We observed full agreement between expert annotations and the corresponding non-expert labels on the sampled subset. For bounding box annotations, minor variations may occur; however, the mean IoU (mIoU) consistently exceeds 0.9, confirming high spatial consistency. These indicate that annotation differences between experts and non-experts are negligible and do not affect the reported performance.
>
> For instructional guide generation (Level 4, Page 5, L254), all Q-A pairs are authored by domain experts, and machine-generated outputs are further reviewed by experts. This is because such tasks require specialized clinical knowledge and domain-specific reasoning, which non-experts cannot reliably provide. Therefore, non-expert annotations are not used for this component. We hope this clarification addresses Reviewer **cug1**’s concern.

---

### Official Review · Reviewer_MgZ9 · 2026-03-12

**Soundness:** 3
**Presentation:** 4
**Significance:** 3
**Originality:** 4
**Overall Recommendation:** 5
**Confidence:** 5

**Summary:**

This paper introduced IVQA-LD, a limb-deficiency–aware benchmark for multimodal understanding and VQA.

**Compliance With Llm Reviewing Policy:**

Affirmed.

**Key Questions For Authors:**

- Do you have any sense of alignment between LLM-as-a-judge scoring and human scores?

**Limitations:**

Yes

**Strengths And Weaknesses:**

Overall, strong paper with a rich dataset in an under researched, but important area.
- Thoughtful task description and thorough metrics
- Clear data generation and preparation pipeline
- Significant improvements to performance with BSI

Paper heavily relies on LLM to generate caption and rehabilitation guide for training. Yes, experts are used to craft captions for eval, but uses LLM to then do eval. I think a correlation analysis with a subset of the results evaluated by humans to ground the LLM-as-a-judge would add value here.

---

> ### Author Rebuttal · Authors · 2026-03-31
>
> We sincerely thank **Reviewer MgZ9** for the highly positive evaluation and constructive feedback. We are greatly encouraged that the reviewer acknowledges our paper as a strong contribution with a rich dataset, thoughtful task descriptions, a clear pipeline, and significant improvements brought by our BSI strategy. We address your question regarding alignment between LLM-as-a-judge and human socres as follows:
>
> >**Q1:** Alignment between LLM-as-a-judge scoring and human scores.
>
> **R1:** Thank you for your constructive suggestion. To evaluate the alignment between LLM-as-a-judge scoring and human scores, we further conduct an expert validation study on 500 randomly sampled questions. Each question is answered by GPT-5, Gemini-2.5-Pro, Qwen2.5-VL Zero-Shot, Qwen2.5-VL SFT, and our Qwen BSI SFT. In the meanwhile, five domain experts (including rehabilitation specialists and biomechanics professionals) are invited to evaluate the generated instructional guides under the Instructional Guide Generation (IGG) setting. The evaluation follows the same 7-dimensional rubric (Figure 8 in Appendix) used in our automated framework, covering accuracy, appropriateness, understandability, organization, actionability, supportive content, and caring tone. Each criterion is scored on a scale from 1 to 5. Table A reports the comparison between human expert ratings and automated LLM-as-a-Judge evaluation scores. We further compute the Pearson correlation coefficient ($r$) to quantify the alignment between expert and LLM evaluations across the 500 samples for each model.
>
> **Table A**: Comparisons between human expert ratings and LLM-as-a-judge scores across different models, along with their Pearson correlation coefficients.
>
> | Model         | Expert Score (Avg) | LLM-as-a-Judge Score (Avg) | Correlation (Pearson $r$) |
> |---------------|--------------------|----------------------------|-------------------------|
> | GPT-5         | 3.77     | 4.03    | 0.82          |
> | Gemini-2.5-Pro| 3.25     | 3.78    | 0.75          |
> | Qwen2.5-VL-7B (Zero-Shot)| 2.05    | 2.43          | 0.88       |
> | Qwen2.5-VL-7B (SFT)      | 4.11    | 4.54          | 0.78       |
> | Qwen2.5-VL-7B (BSI SFT)  | 4.28    | 4.64          | 0.84       |
>
> As shown in Table A, the overall trend and model performance rankings remain highly consistent. This demonstrates a strong correlation between human and automated evaluations, indicating that the LLM-as-a-Judge framework provides a reliable and consistent proxy for expert assessment. We will include this alignment experiment and discussion in the revision.

---

> > ### Author Rebuttal · Reviewer_MgZ9 · 2026-04-07
> >
> > Thank you for thoroughly addressing my comments.

---

### Official Review · Reviewer_7KCp · 2026-03-19

**Soundness:** 3
**Presentation:** 4
**Significance:** 4
**Originality:** 4
**Overall Recommendation:** 4
**Confidence:** 4

**Summary:**

The paper introduces IVQA-LD, the first large-scale, expert-annotated dataset designed to address the critical exclusion of people with limb differences from current AI services and vision-language models. The motivation stems from the discovery that state-of-the-art models often fail to detect missing limbs or accurately reason about prosthetic devices, leading to systematic errors and inequitable model behaviors. To bridge this gap, the authors curated 80,000 VQA pairs from 14,054 images, integrating medical-functional classes defined by the World Health Organization (WHO) and World Para Athletics (WPA) to ensure clinical relevance. Their primary methodological contribution is the Body-centric Structure-aware Initialization (BSI) strategy, a two-stage fine-tuning curriculum that first aligns models with limb-specific structural priors—such as residual limb locations—before supervising them on joint localization and reasoning. This approach dramatically improves performance across eight core tasks, ranging from visual grounding to rehabilitation guide generation, allowing adapted models to internalize disability-specific semantics that general-purpose models typically fail to encode.

**Compliance With Llm Reviewing Policy:**

Affirmed.

**Key Questions For Authors:**

See weaknesses

**Limitations:**

See weaknesses

**Strengths And Weaknesses:**

Strengths:
- This paper addresses a critical gap in AI fairness by introducing IVQA-LD, the first large-scale, expert-annotated dataset for individuals with limb deficiency. Existing models often fail to recognize prosthetic devices or "hallucinate" intact limbs because this population is underrepresented in standard training data. By providing 80,000 VQA pairs derived from diverse real-world and Paralympic scenarios, the authors create a high-quality benchmark that directly improves model representation and inclusivity.
- A major strength is the dataset’s integration of professional medical and athletic standards. The authors collaborated with rehabilitation and biomechanics experts to ensure annotations align with WHO ICF functional classes and World Para Athletics (WPA) categories.
- The paper is well written and easy to follow.


Weaknesses:
- The paper’s evaluation of generative tasks, particularly the "Instructional Guide Generation (IGG)," relies heavily on an LLM-as-a-Judge framework using GPT-4.1-mini. Although the authors utilize a structured rubric assessing seven clinical dimensions—such as accuracy, actionability, and supportive content—this automated approach can be subjective and prone to the inherent biases of large language models.
- Another weakness involves the relatively low audit rate for the human-authored portions of the dataset. While domain experts were involved in designing the tasks, the formal quality control protocol only subjected 5% of the human-authored Q-A pairs (roughly 125 samples) to a secondary expert cross-review. For a specialized dataset containing 80,000 pairs, this small sampling size may not be enough to eliminate all subtle errors or inconsistencies.

---

> ### Author Rebuttal · Authors · 2026-03-31
>
> We sincerely thank **Reviewer 7KCp** for the appreciative and constructive comments. We are glad that the reviewer recognizes our paper addresses a critical gap in AI fairness, provides a high-quality benchmark that integrates professional medical and athletic standards, and is well-written. We address the concerns raised by the reviewer below.
>
> >**Q1:** Reliance on an LLM-as-a-Judge framework using GPT-4.1-mini for Instructional Guide Generation (IGG) evaluation may introduce subjectivity and LLM biases.
>
> **R1:** Thank you for your insightful comment. To assess the potential subjectivity and bias of the LLM-as-a-Judge framework, we further conduct an expert validation study on 500 randomly sampled questions. Each question is answered by GPT-5, Gemini-2.5-Pro, Qwen2.5-VL Zero-Shot, Qwen2.5-VL SFT, and our Qwen BSI SFT. Then, five domain experts (including rehabilitation specialists and biomechanics professionals) are invited to evaluate the generated instructional guides under the Instructional Guide Generation (IGG) setting. The evaluation follows the same 7-dimensional rubric used in our automated framework (Figure 8 in Appendix), covering accuracy, appropriateness, understandability, organization, actionability, supportive content, and caring tone. Each criterion is scored on a scale from 1 to 5. Table A reports the comparison between human expert ratings and automated LLM-as-a-Judge evaluation scores. We further compute the Pearson correlation coefficient ($r$) to quantify the alignment between expert and LLM evaluations across the 500 samples for each model.
>
>
> **Table A**: Comparisons between human expert ratings and LLM-as-a-judge scores across different models, along with their Pearson correlation coefficients.
>
> | Model         | Expert Score (Avg) | LLM-as-a-Judge Score (Avg) | Correlation (Pearson $r$) |
> |---------------|--------------------|----------------------------|-------------------------|
> | GPT-5         | 3.77     | 4.03             | 0.82          |
> | Gemini-2.5-Pro| 3.25     | 3.78             | 0.75          |
> | Qwen2.5-VL-7B (Zero-Shot)| 2.05    | 2.43             | 0.88          |
> | Qwen2.5-VL-7B (SFT)      | 4.11    | 4.54             | 0.78          |
> | Qwen2.5-VL-7B (BSI SFT)  | 4.28    | 4.64             | 0.84          |
>
> As shown in Table A, the overall trend and model performance rankings remain highly consistent. This demonstrates a strong correlation between human and automated evaluations, indicating that the LLM-as-a-Judge framework provides a reliable and consistent proxy for expert assessment. We will include these validation results and a detailed discussion in the revision.
>
> >**Q2:** The 5% audit rate may be insufficient to eliminate subtle errors in an 80,000-pair dataset.
>
> **R2:** We would like to clarify that the “5% of the human-authored Q–A pairs (approximately 125)” reported on Page 5 (L259, right column) **does not refer to the entire 80,000-pair dataset**. Instead, it applies only to the subset of **expert-generated captions and guidance** (also used for testing), which consists of 2,500 Q–A pairs in total; 5% of this subset corresponds to 125 samples.
>
> Since these Q–A pairs are **carefully curated by domain experts**, we adopt a **sampling-based audit strategy (5%)** for quality assurance. In contrast, **all machine-generated Q–A pairs**—including those for image captions and rehabilitation guidance—are fully reviewed by experts to ensure reliability, while also balancing expert effort and time.
>
> To further address the reviewer’s concern, we conducted an additional expert cross-review on an extra 10% of the human-authored subset. The review confirms that the newly selected 250 Q–A pairs are correct. This suggests that annotation noise is minimal and unlikely to affect the reported results. We will clarify this in the revised version.

---

> > ### Author Rebuttal · Reviewer_7KCp · 2026-04-07
> >
> > My issues have been addressed.

---

### Decision · Program_Chairs · 2026-04-30

**Decision:**

Accept (regular)

**Comment:**

This paper introduces IVQA-LD, a large-scale multimodal VQA benchmark designed to address the exclusion of individuals with limb differences in current vision-language models, alongside a Body-centric Structure-aware Initialization (BSI) strategy. All reviewers thought the paper for tackling an under-explored problem in AI fairness, noting the dataset's rich, expert-informed curation using WHO and Paralympic standards, and the empirical effectiveness of the BSI approach. Initial reviewer concerns centered the reliability of using an LLM-as-a-judge for generative tasks, the proportion and auditing of non-expert annotations, and missing methodological details regarding data splits and the grounding pipeline. In rebuttal, the authors addressed these issues by providing a human-expert correlation study validating the LLM judge, clarifying the quality control and task-specific division for non-expert annotations, confirming a video-level data splitting protocol to prevent leakage, and supplying ablation studies. As all reviewers acknowledged that their concerns were resolved, this paper is recommened for acceptance.